

# Varying chiral ratio of Pinic acid enantiomers above the Amazon rainforest

Denis Leppla[1], Nora Zannoni[2], Leslie Kremper[2], Jonathan Williams[2], Christopher Pöhlker[2], Marta Sá[3], Maria Christina Solci[4], Thorsten Hoffmann[1]

[1]Chemistry Department, Johannes Gutenberg-University, Mainz, 55128, Germany
[2]Max Planck Institute for Chemistry, Mainz, 55128, Germany
[3]Instituto Nacional de Resquisas da Amazônia/INPA, Manaus/AM, Brazil
[4]Universidade Estadual de Londrina, Londrina/PR, Brazil

*Correspondence to*: Thorsten Hoffmann (hoffmant@uni-mainz.de)

**Abstract.** Chiral chemodiversity plays a crucial role in biochemical processes such as insect and plant communication. However, the vast majority of organic aerosol studies do not distinguish between enantiomeric compounds in the particle phase. Here we report chirally specified measurements of secondary organic aerosol (SOA) at the Amazon Tall Tower Observatory (ATTO) at different altitudes during three measurement campaigns at different seasons. Analysis of filter samples by liquid chromatography coupled to mass spectrometry (LC-MS) has shown that the chiral ratio of pinic acid ($C_9H_{14}O_4$) varies with increasing height above the canopy. A similar trend was recently observed for the gas-phase precursor α-pinene, but more pronounced. Nevertheless, the measurements indicate that neither the oxidation of (+/−)-α-pinene nor the incorporation of the products into the particulate phase proceeds with stereo preference and that the chiral information of the precursor molecule is merely transferred to the low-volatility product. The observation of the weaker height gradient of the present enantiomers in the particle phase at the observation site can be explained by the significant differences in the atmospheric lifetimes of reactant and product. Therefore, it is suggested that the chiral ratio of pinic acid is mainly determined by large-scale emission processes of the two precursors, while meteorological, chemical, or physicochemical processes do not play a particular role. Characteristic emissions of the chiral aerosol precursors from different forest ecosystems, in some cases even with contributions from forest related fauna, could thus provide large-scale information on the different contributions to biogenic secondary aerosols via the analytics of the chiral particle-bound degradation products.

## 1 Introduction

The Amazon Basin is one of the most pristine ecosystems on Earth and extends over a region of about 6 million $km^2$ in South America. Approximately 80% of this region is covered with rainforest which accounts for circa 40% of all tropical forests on the globe (Goulding and Barthem, 2003; Andreae et al., 2015). Various studies estimated the total aboveground carbon content of the Amazon forest at about 86 – 108 Pg C (Saatchi et al., 2007; Baccini et al., 2012), demonstrating its crucial role in climate change and life on Earth in general. In addition to the importance of the Amazon rainforest for the





global carbon cycle, its role within the hydrological cycle is one of its most important functions and responsible for the most biodiverse ecosystem on Earth (Hoorn et al., 2010; Wittmann et al., 2013). The immense biodiversity, however, is constantly threatened by deforestation and land-use change (Vieira et al., 2008; Pöhlker et al., 2019). In particular, forest fires are the primary source of pollution. One of the consequences of these anthropogenic activities are significant seasonal variations in

aerosol particle concentrations in the Amazon basin, which strongly affect the radiation budget, cloud physics, and precipitation (Martin et al., 2010; Pöschl et al., 2010; Artaxo et al., 2013). However, even without human influence an estimated 760 Tg C is released into the atmosphere annually worldwide in the form of biogenic volatile organic compounds (BVOCs) (Sindelarova et al., 2014). Rapid reactions with atmospheric reactants, such as OH radicals and ozone, lead to products with lower volatility and subsequently to the formation of secondary organic aerosols (SOAs) (Kroll and Seinfeld,

2008). Organic compounds account for up to 90% of total atmospheric particulate matter in tropical forests (Kanakidou et al., 2004), and directly and indirectly affect the Earth's climate.

Monoterpenes are considered an important fraction of BVOCs comprising a huge variety of individual chemical compounds (Guenther et al., 2012). Many of those are chiral and carry stereogenic centers, resulting in enantiomeric structures that cannot be superimposed. Although chiral molecules play a critical role in insect and plant communication (Phillips et al.,

2003; Mori, 2014), they are rarely distinguished in studies regarding atmospheric chemistry. The reason for this is that enantiomers have identical physical and chemical properties in an otherwise non-chiral environment, such as boiling point, exact mass but also reaction rate with atmospheric oxidants (OH, $O_3$, and $NO_3$). However, when oligomerization based on chiral building blocks occurs, a process also relevant to secondary organic aerosol formation (Tolocka et al., 2004; Hallquist et al., 2009), changes in physicochemical properties may well occur. It has been shown that physical and chemical

properties, such as melting point and water solubility, can then be determined by stereochemistry (Katsumoto et al., 2010; Baker et al., 2015; Cash et al., 2016). Thus, chirality might affect the ability of SOA formation and, consequently, influence for example the radiative forcing and cloud processing of aerosol particles.

Recently, Zannoni et al. (2020) have reported significant variances of the enantiomeric ratio of α-pinene with increasing altitude at the ATTO site in the Amazon rainforest. This unexpected result was attributed to strong local sources, such as

insects, which may be responsible of changing the predominant ambient ratio of (−)- and (+)-α-pinene. Through subsequent atmospheric oxidation, the chiral information should be transferred to the particle phase since for certain oxidation reactions the stereochemical centers are retained. Pinic acid is an oxygenated product of α-pinene ozonolysis and a key product of atmospheric VOC degradation compounds identified at the molecular level (Christoffersen et al., 1998; Ma et al., 2007; Yasmeen et al., 2011; Mutzel et al., 2016). Due to its two carboxyl functions, it has a relatively low vapor pressure and thus

a strong tendency to partition into the particle phase. Another important product of the α-pinene oxidation is 3-methyl-1,2,3-butanetricarboxylic acid (MBTCA) (Szmigielski et al., 2007; Yasmeen et al., 2010; Müller et al., 2012; Claeys et al., 2007), which would actually be a better marker substance for particle-bound VOC oxidation products due to its even lower volatility. However, MBTCA has lost the original chiral information due to the breakup of the four-membered ring. Here we



show that the chiral ratio of pinic acid in the Amazon rainforest exhibits an excess of one enantiomer. The results reflect the
trend observed in the gas phase indicating a direct link between BVOC emissions and SOA concentrations through chirality.

## 2 Experimental section

### 2.1 Measurement campaigns and filter sampling

Ambient aerosol particles were collected during three campaigns (10/22/2018 – 10/31/2018, 03/04/2019 – 03/14/2019,
09/20/2019 – 09/26/2019) in the Amazon rainforest at the Amazon Tall Tower Observatory (ATTO) station. The research
site is already well described by Andreae et al. (2015). Briefly, ATTO is located about 150 km northeast of Manaus, Brazil,
and about 12 km northeast of the Uatumã River within the Uatumã Sustainable Development Reserve (UDSR). The site is
located on a plateau ~120 m above sea level and is surrounded by dense, non-flooded upland forest (terra firme) with a
maximum tree canopy height of approximately 40 m. The area between the river and the plateau is primarily characterized
by floodplain forests and white-sandy soil ecosystems of the savanna (campinas) and forest (campinarana) types. The
seasonal shift of the Intertropical Convergence Zone (ITCZ) influences meteorological characteristics at the ATTO with
clean and wet conditions from February to May and polluted and dry conditions from August to November (Pöhlker et al.,
2019) (more details can be found in the SI). Key meteorological parameters, trace gas concentrations, and particle
concentrations are continuously recorded at the station.

PM2.5 filter samples were collected from the 80 m, 150 m, and 320 m platforms of the ATTO tower (coordinates: S
02°08.752' W 59°00.335') during the 2018 dry season and 2019 wet season. During the 2019 dry season, an instrument was
moved from the 150 m platform to the ground. Aerosol particles were collected on filters consisting of borosilicate glass
microfibers bonded with PTFE (Pallflex® Emfab, 70 mm diameter). To exclude particles with aerodynamic diameters larger
than 2.5 µm, a pre-separator (Digitel, DPM2.3) was installed upstream of the filter holder. The flow rate was kept constant at
38 L min$^{-1}$. In order to distinguish between day and nighttime, the sampling time was 12 h on average. This also ensured
85  sufficient aerosol mass on the filter. Additionally, particle size resolved filter samples were collected at the smaller 80 m
triangular mast (coordinates: S 02°08.602' W 59°00.033'). For this purpose, an aerosol inlet at a height of 60 m was
connected to a micro-orifice uniform deposit impactor (MOUDI) (TSI, 125R), which allowed sampling of 13 different stages
with particle diameters between 10 µm and 10 nm. The flow rate was kept constant at 10 L min$^{-1}$. The filter samples were
stored in a freezer at – 25 °C until analysis. A total of 203 filter samples were collected and analyzed.

90  ### 2.2 Chamber experiments

We performed ozonolysis experiments with enantiomerically pure (−)- and (+)-α-pinene (Sigma Aldrich, 99%, optical purity
ee: 97%) to identify the resulting pinic acid enantiomers. For this purpose, we used a 100 L glass reaction chamber that was
completely darkened. Prior to an experiment, the chamber was purged with synthetic air overnight to remove residual
particles and reactants from the gas phase. In addition, an activated carbon trap was used to eliminate all organic compounds





95    in the air stream. For each experiment, the chamber was flushed with humidified synthetic air at a constant rate of 6 L min$^{-1}$,

resulting in a relative humidity of 50%. Ozone was generated using the calibration system of an ozone analyzer (Dasibi,

1008-RS) and then introduced into the chamber at a flow rate of 3 L min$^{-1}$. The ozone concentration stabilized at about

800 ppb. The VOC precursor compound was placed in a diffusion test gas source maintained at 40 °C (Thorenz et al., 2012).

A flow rate of 1 L min$^{-1}$ was set to transport the VOCs into the reaction chamber. On average, an α-pinene concentration of

200 ppb was present in the chamber. The particle size distribution for diameters between 10 – 800 nm was recorded using a

Scanning Mobility Particle Sizer (SMPS) (GRIMM, Advanced CPC 5.416). Details can be found in the SI.

## 2.3 Sample preparation and LC-MS analysis

One half of each filter was extracted three times with 1.5 mL in a 9:1 mixture of methanol and water (Fisher Scientific,

Optima™ grade) on a laboratory shaker for 30 min. The resulting extracts were combined and filtered through 0.2 µm PTFE

syringe filters (Carl Roth, Rotilabo® KC94.1) to remove undissolved material. The solvent was then completely evaporated

under a gentle N$_2$ stream and 700 µL of a 9:1 mixture of water and acetonitrile (Fisher Scientific, Optima™ grade) was

added to the remaining residue. All samples were analyzed using an ultra-high performance liquid chromatograph (UHPLC)

(ThermoFisher Scientific, UltiMate 3000) coupled to an Orbitrap mass spectrometer (MS) (ThermoFisher Scientific, Q

Exactive™). The instrument was equipped with an electrospray ionization (ESI) source and operated in the negative mode.

The MS system was regularly calibrated with a negative ion calibration solution (Fisher Scientific, Pierce™). The mass

resolution was 140 000 at *m/z* 200 with mass accuracies less than 1 ppm. To obtain sufficient ion signal intensities, the

following ESI-MS parameters were applied: Spray voltage −3.2 kV; capillary temperature 300 °C; auxiliary gas flow 10;

sheath gas flow 30; S-lens RF level 50%. Chiral separation was performed on an amylose-based tris(3-chloro-5-

methylphenylcarbamate) column (150 × 2.1 mm ID, 5 µm, Daicel, CHIRALPAK® IG). Eluent A (water with 2%

acetonitrile and 0.04% formic acid) and eluent B (acetonitrile with 2% water) were used for isocratic separation (80:20) at a

constant flow of 200 µL min$^{-1}$. The column temperature was maintained at 25 °C. To ensure the identification of pinic acid

in the collected aerosol samples, a standard compound was synthesized according to Moglioni et al. (2000). Seven-point

calibration curves were generated with concentrations of pinic acid ranging from 0.5 ng mL$^{-1}$ to 500 ng mL$^{-1}$.

## 3 Results and discussion

To identify the two enantiomers of pinic acid, a synthesized reference standard was analyzed by HPLC-MS. The LC method

was optimized to ensure baseline separation of the two stereoisomers. The corresponding chromatogram is shown in Fig. 1,

upper panel (a). Integration of the reference standard resulted in almost equal signal areas, indicating a racemic standard

compound. As confirmed by chamber experiments (lower panel, (c)), the first signal (red line) can be assigned to the

oxidation product of enantiomerically pure (+)-α-pinene. It is referred to as E1 in this study. In the subsequent experiment,

enantiomerically pure (−)-α-pinene was oxidized. Accordingly, the later eluting second signal resulted from the ozonolysis





of pure (−)-α-pinene and is designated E2. As can be seen in the bottom panel (c) of Fig. 1, E1 was also detected after oxidation of (−)-α-pinene. This was caused by memory effects from the previous chamber experiment with (+)-α-pinene, a common observation in SOA experiments caused by pinic acid condensation on the chamber walls and revolatilization in subsequent experiments. This was also confirmed by the observation of a steady decrease in this carryover in further

experiments. However, these results allowed the unambiguous identification and quantification of both pinic acid stereoisomers in the measurements of filter samples from the Amazon rainforest. Since the particular molecular formula $C_9H_{14}O_4$ (*m/z* 185.0819) is common for oxidation products of various monoterpenes (Kourtchev et al., 2015), for example also for the products of the oxidation of Δ-3-carene and sabinene, i.e. sabinic and 3-caric acid (Larsen et al., 2001), additional signals were observed in the measurements at the ATTO tower besides those formed by the α-pinene oxidation

(middle panel (b)). No further assignment of these signals was made in this study.

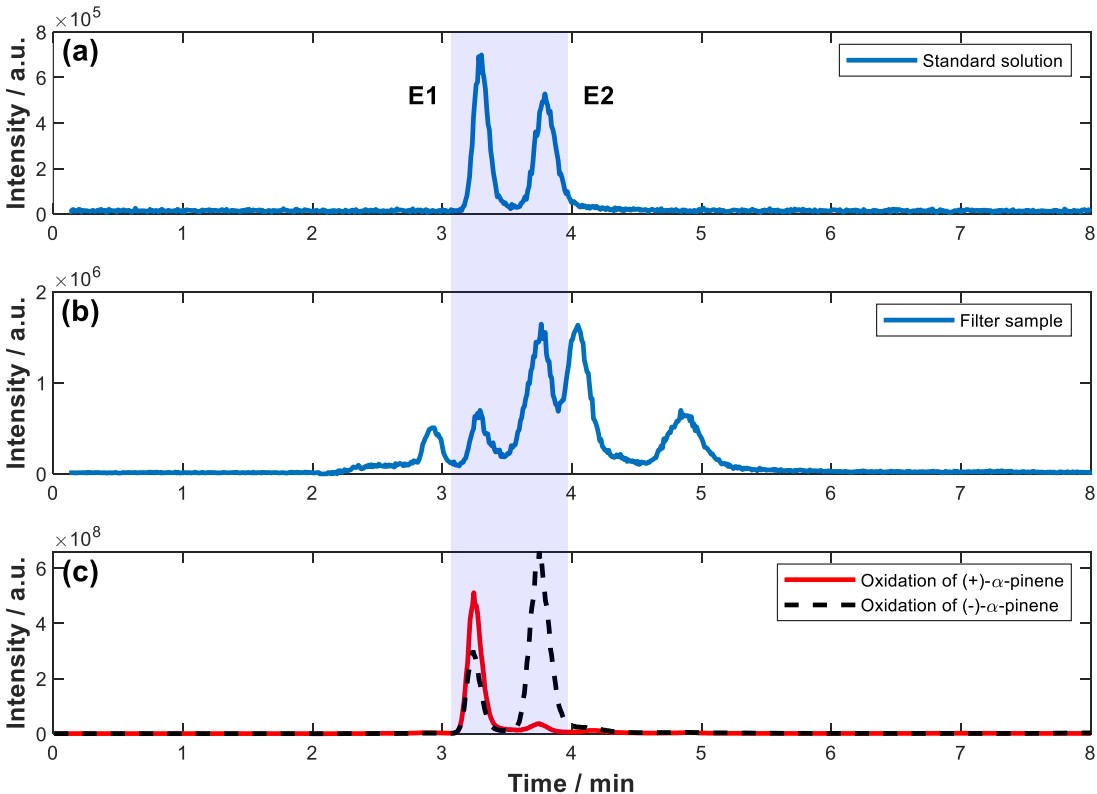

**Figure 1: Typical chromatograms of pinic acid $C_9H_{14}O_4$ ([M−H]⁻ *m/z* 185.0819). Both stereoisomers are baseline separated in a reference standard aqueous solution (a). Integration resulted in equal areas, which indicates a racemic mixture. The middle panel (b) illustrates an exemplary analysis of a filter sample. These experiments revealed additional signals at *m/z* 185.0819, a typical**
**mass for monoterpene oxidation products. The lower panel (c) highlights the results of the chamber experiments to distinguish between both stereoisomers of pinic acid (E1 in red results from the ozonolysis of (+)-α-pinene, E2 in dashed black results from the ozonolysis of (−)-α-pinene).**



### 3.1 Vertical concentration gradients of Pinic acid on PM2.5 filter samples

Zannoni et al. (2020) showed with measurements made at the ATTO tower that the enantiomeric ratio of α-pinene varies
with (+)-α-pinene dominating at canopy level and (−)-α-pinene at the top level of the tower. The ratio was observed to be
independent of wind direction and speed and the authors suggest the presence of a potent uncharacterized local (+)-α-pinene
rich source, possibly linked to herbivory and termites. Here, we report similar observations for the chiral ratio of pinic acid
enantiomers in the particle phase. Figure 2 illustrates the atmospheric concentrations of both stereoisomers in ng m$^{-3}$
measured at different heights at ATTO during the dry season in October 2018 (denoted as DS18), the wet season in March
2019 (WS19), and the dry season in September 2019 (DS19). The displayed data correspond to the calculated mean values
for the respective campaign in the morning hours (07:00 – 12:00 local time only for DS19), during daytime (07:00 – 17:00
local time for DS18 and WS 19, 12:00 – 17:00 local time for DS19), and during nighttime (17:00 – 07:00 local time). The
instrumental detection limits were sufficiently low for both pinic acid stereoisomers (see Fig. S5), which enabled the reliable
quantification of most atmospheric concentrations. However, some samples revealed concentrations of E1 below the
detection limit. This occurred mainly for samples taken during the wet season with generally lower particle concentrations
(see SI) and for measurements at 320 m during DS19. Nevertheless, these values were included in the calculation of the
chiral ratio, as they still represent the trend.

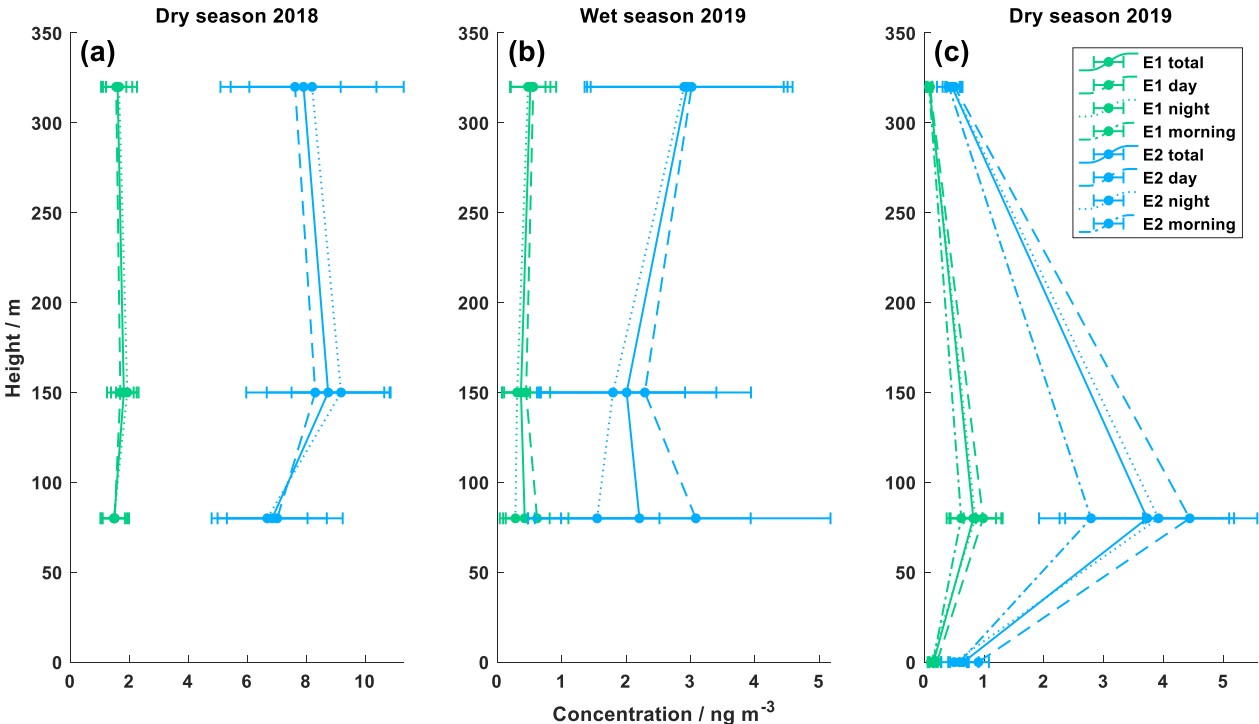

**Figure 2: Vertical concentration profiles of the pinic acid enantiomers E1 and E2 during three different campaigns ((a) dry season**
**2018 = October 2018, (b) wet season 2019 = March 2019, (c) dry season 2019 = September 2019). Aerosol filter samples were**
**collected at 80 m, 150 m, and 320 m for the Dry season 2018 and the Wet season 2019. For the Dry season 2019, the instrument at**





**150 m was moved to the ground level at 0 m. Daytime refers to 7:00 – 17:00, nighttime refers to 17:00 – 7:00 during the first two campaigns. For the third campaign, the time resolution was improved with morning time between 7:00 – 12:00, daytime between 12:00 – 17:00, and nighttime between 17:00 – 7:00. All times are local times. The data points correspond to the mean values of all**
**measurements.**

As can be seen in Fig. 2, all samples had higher concentrations of E2 than E1 with maximum values at 80 m and 150 m for the dry seasons. The higher concentrations of the E2 precursor (−)-α-pinene could be an explanation for this observation. The highest average concentrations of E2 were measured during DS18 at 150 m with $(9 \pm 2)$ ng m$^{-3}$ (errors are represented by one standard deviation of the data sets), while the lowest average concentrations of $(0.09 \pm 0.04)$ ng m$^{-3}$ were recorded

during DS19 at 320 m. All values are listed in Table 1. Contrary to expectations, the concentrations at 320 m during DS19 were significantly lower than during DS18 and below the values of WS19. However, during this period (DS19), significantly lower total particle masses were also observed (see SMPS data in Fig. S8). This may have caused the significantly lower pinic acid concentrations on the top of the sampling platform. During the DS19 campaign, the aerosol collector was moved from 150 m to the ground level to include chemistry below the canopy in our observations. Concentrations for pinic acid

were lower below the canopy, which was also reported by Plewka et al. (2006) during measurements in a coniferous forest in Germany. They concluded that increased actinic radiation above the canopy could be the reason for this observation. However, the significantly lower ozone concentrations below the canopy and particle sinks due to dry deposition also fit this observation. Thus, the measured concentration profiles suggest photochemical processes and oxidation chemistry above the forest. Additionally, as recently reported by Bourtsoukidis et al. (2018), soil can be excluded as a potentially effective

monoterpene source at ATTO. Consequently, lower ground level concentrations of pinic acid are expected. The α-pinene emissions exhibit a pronounced day-night cycle and reach their maximum together with photosynthetically active radiation (PAR) and temperature at about 11:00 to 14:00 local time (Zannoni et al., 2020). In comparison, a temporal shift of the diel concentration maxima is expected for the particulate phase, caused by mixing and chemical aging. As assumed, the mean concentrations of the measured VOC oxidation products were higher during the daytime in each campaign, except for 80 m

and 320 m during DS18 (Fig. 2). But, of course, the coarse temporal resolution of the present measurements prevents the determination of more detailed diel concentration profiles.

**Table 1: Average concentration values for E1 and E2 at different heights during three measurement campaigns. Errors are represented as standard deviations of the dataset. Additionally, the calculated enantiomeric ratio, ER, is listed.**

| | Height / m | avg. c(E1) / ng m$^{-3}$ | | avg. c(E2) / ng m$^{-3}$ | | avg. ER (E2 / E1) | |
|---|---|---|---|---|---|---|---|
| **DS18** | 80 | 1.48 | ± 0.43 | 6.84 | ± 1.85 | 4.62 | ± 0.11 |
| | 150 | 1.81 | ± 0.43 | 8.74 | ± 2.08 | 4.84 | ± 0.10 |
| | 320 | 1.60 | ± 0.49 | 7.91 | ± 2.47 | 4.95 | ± 0.12 |
| **WS19** | 80 | 0.42 | ± 0.39 | 2.21 | ± 1.73 | 5.20 | ± 0.19 |
| | 150 | 0.37 | ± 0.30 | 2.01 | ± 1.39 | 5.48 | ± 0.15 |
| | 320 | 0.51 | ± 0.31 | 2.95 | ± 1.56 | 5.77 | ± 0.12 |
| **DS19** | 0 | 0.15 | ± 0.07 | 0.65 | ± 0.25 | 4.23 | ± 0.11 |
| | 80 | 0.82 | ± 0.38 | 3.72 | ± 1.37 | 4.54 | ± 0.12 |
| | 320 | 0.09 | ± 0.04 | 0.46 | ± 0.16 | 5.30 | ± 0.09 |





Figure 3 shows the calculated enantiomer ratios for both stereoisomers of pinic acid, E2 / E1. The most remarkable result is

that the ratio increases consistently with tower height over all measurement periods. Although the absolute concentrations of the pinic acid enantiomers vary considerably between campaigns, the chiral ratios are always in the range 4 – 6. The dry seasons are characterized by ratios of about $4.6 \pm 0.1$ and $4.5 \pm 0.1$ at 80 m, while the wet season shows elevated values of $5.2 \pm 0.2$. At 320 m, the E2 enantiomer resulting from (−)-α-pinene oxidation is even more dominant, leading to ratios of $4.9 \pm 0.1$ and $5.30 \pm 0.09$ during the dry seasons and $5.8 \pm 0.1$ during the wet season, respectively. Compared to the vertical

concentration gradients of pinic acid in Fig. 2, the enantiomer ratios increase with tower height regardless of the absolute concentrations. This fits very well with the observations of Zannoni et al. (2020), as they described similar results for the chiral ratio of (−)-α-pinene / (+)-α-pinene, i.e., the precursors of the analytes measured here. Increasing values from 0.38 to 6.5 (40 m and 320 m, respectively) were reported, with (−)-α-pinene being the dominant species in the gas phase at the top of ATTO. The chamber experiments performed (Fig. 1) showed that E2 is the oxidation product of (−)-α-pinene. Therefore, as

shown in Fig. 3, the chiral ratio in the particulate phase behaves similarly to that in the gas phase, but much less pronounced. It ranges from $4.2 \pm 0.1$ near the ground to $5.8 \pm 0.1$ at 320 m altitude. Remarkably, as reported by Zannoni et al. (2020), there was little variation in the two α-pinene enantiomers in the gas phase during the 2018 wet season. This finding is not consistent with the results from the particulate phase as shown in Fig. 3, at least when compared to the 2019 results. Similar to the dry seasons, the chiral ratio of pinic acid also shows steady growth with increasing tower height during the wet season.

The emission of α-pinene and the ratio of enantiomers released is determined by the vegetation and thus, for ecosystem measurement sites such as ATTO, by the local flora, possibly with additional contributions from the local fauna, such as termites (Zannoni et al., 2020). Measurements at different altitudes in the present study, thus first reflect differences in releases from the different source regions, which, after all, depend on the effective measurement altitude (Arriga et al., 2017). Thereby, the measurements reflect local sources at the lower levels and regional influences at the upper levels

(Bakwin et al., 1998; Andrews et al., 2014). Due to the high reactivity of biogenic VOCs toward OH radicals and ozone, the mean tropospheric lifetime of α-pinene is on average only 2.6 h and 4.6 h, respectively (Atkinson, 1997, 2000; Kesselmeier and Staudt, 1999). Thus, strong local sources such as insects can alter the enantiomeric distributions in the gas phase. Consequently, the chiral ratio for α-pinene at ATTO varies strongly with increasing tower height. However, the oxidative formation and especially depletion of pinic acid occur on different time scales with estimated lifetimes of several days

(Rogge et al., 1993; Rudich et al., 2007; Schauer et al., 1996). The difference in the two enantiomers is therefore much less pronounced for the chiral oxidation products, but also reflects the different footprint regions.





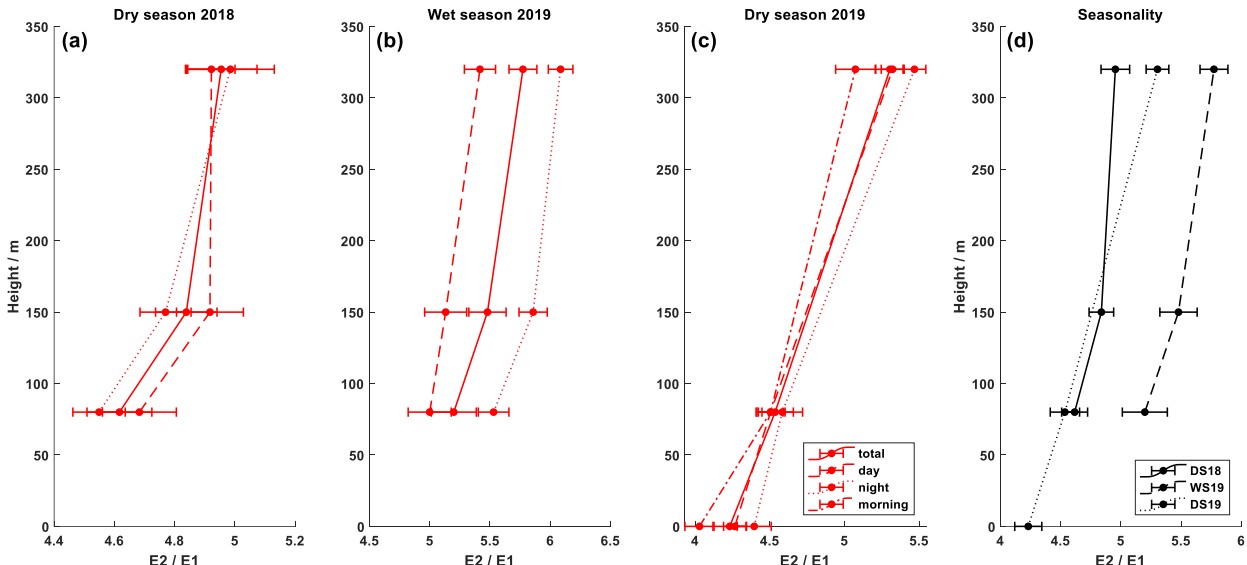

**Figure 3: Vertical profiles of the enantiomeric ratio E2 / E1 (a – c). The local times correspond to those shown in Fig. 2. Additionally, the comparison for all campaigns is shown in panel (d) (DS = dry season, WS = wet season). There is a constant**
**increase of E2 compared to E1 with increasing altitude. All values are in the same magnitude. However, the wet season was characterized by slightly larger ratios.**

### 3.2 Daily variations of Pinic acid concentrations

Figure 4 shows the absolute concentrations for E1 and E2 of pinic acid near the ground (blue), at 80 m (dark green), 150 m (green), and 320 m (light green) for the three measurement campaigns. In general, absolute concentrations at the respective
tower heights vary rather slightly within each campaign, with the greatest variation just above the canopy on the 80 m platform. It is clear that the 80 m values are more influenced by local sources than the 150 m and 320 m heights. This is evident for both the DS19 and WS19 campaigns. In particular, the daytime samples (03/06/2019, 03/07/2019, and 03/09/2019) show different concentration trends at 80 m height compared to the higher tower levels. An obvious explanation is characterized by an effective coupling between the forest canopy and the superjacent atmosphere caused by strong
turbulent activity during the daytime (Andreae et al., 2015). Measurements within this roughness sublayer (RSL) are expected to be strongly influenced by the various roughness elements. In general, the RSL is thought to extend to approximately 2 to 3 times the canopy height of about 40 m (Raupach et al., 1996; Williams et al., 2007; Chor et al., 2017). In addition, according to Dias-Júnior et al. (2019), the RSL merges directly into the adjacent convective mixed layer. Thus, similar conditions exist for both 150 m and 320 m heights, resulting in decoupled concentration profiles compared to 80 m.
The proportions of the two enantiomers showed almost constant ratios both for the different sampling heights and within each measurement campaign (see Fig. 4b, d, f). However, as indicated above, the ratios were also quite similar between the different campaigns. A certain trend can be seen in the dry season of 2019 (Fig. 4f), as here the relative concentration of E2



increased over time at all observation heights. All these observations supported the idea that the chiral ratio is determined by mixing and transport processes rather than by the preferential formation of a particular enantiomer.

**Figure 4: Absolute concentrations of E1 and E2 at different heights at ATTO are displayed in (a), (c), and (e) for three different measurement campaigns, respectively. The resulting chiral relative abundance of E2 is illustrated in (b), (d), and (f). Each data point corresponds to a sample collected at ATTO. The analysis of aerosol samples at ground level (0 m) was only performed during the dry season 2019.**

## 3.3 Particle size resolved measurements at the triangular mast

In addition to the PM2.5 filter samples, MOUDI measurements were carried out to investigate the chiral ratio of pinic acid at different particle diameters. Size resolved samples were collected at the 60 m inlet of the triangular mast during each campaign. Figure 5 shows the concentration of E1 and E2 in the different particle fractions. Similar to the PM2.5 samples, all MOUDI samples were dominated by the E2 enantiomer. However, especially in the size range with small particles (diameter ≤ 100 nm) and large particles (diameter ≥ 1000 nm), low total particle mass concentrations were measured (see Fig. S9), resulting partly in concentrations of the dicarboxylic acid below the detection limit. Nevertheless, these values were





included in the figure as they illustrate the general trend. All measurements showed elevated concentrations of pinic acid for the particle size range between 180 nm and 560 nm diameter. Maximum concentrations of $(0.027 \pm 0.002)$ ng m$^{-3}$ for E1 and $(0.113 \pm 0.023)$ ng m$^{-3}$ for E2 were determined in both dry seasons. The rainy season reached maximum concentrations of

$(0.026 \pm 0.020)$ ng m$^{-3}$ for E1 and $(0.077 \pm 0.059)$ ng m$^{-3}$ for E2. It should be noted that these results are mean concentrations that include four and two MOUDI measurements, respectively. This is due to the fact that the sampling time was significantly extended to ensure sufficient aerosol mass loadings on the filters.

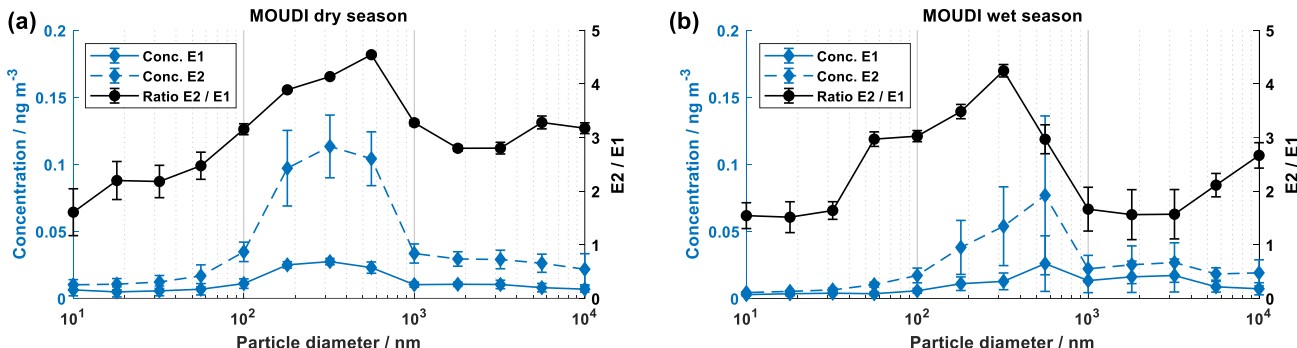

**Figure 5: Particle size resolved MOUDI measurements of pinic acid at the triangular mast (60 m) (dry season = (a), wet season =**
**(b)). The concentrations of E1 and E2 are illustrated together with the resulting chiral ratio. Both seasons revealed elevated concentrations in the particle size range between 180 – 560 nm diameters. The concentrations for small particle diameters (< 56 nm) were partially below the limit of detection. The respective values are still shown to illustrate the trend. The data points correspond to the mean values of all measurements.**

The concentrations of E1 and E2 showed essentially similar size distributions as the particle size distribution measured with

SMPS systems during the respective measurement periods (see Fig. S9). Accordingly, the highest concentrations of pinic acid are found in particles with diameters between 100 nm and 600 nm, which represents the accumulation mode of the particle size distribution. In fact, the absolute concentrations are slightly lower compared to the samples collected at ATTO, which could have two reasons. First, the MOUDI samples were collected with a 60 m inlet, which may result in particle losses in the inlet lines. Second, the sampling time was extended to several days, which could obviously favor sampling

artifacts such as evaporation/decomposition of the target analyte during sampling. Consequently, lower concentrations of pinic acid are found. The resulting chiral ratios between E2 and E1 tend to be lower, with average values of 4.2 and 3.6 for the dry and wet seasons, respectively, but still quite similar to the measurements presented previously at ATTO. The observed shape of the size distribution of the ratio of the two enantiomers, also shown in Fig. 5, is interesting but not easily explained. The comparatively high relative contributions of E1 (E2 / E1 value ratios between 1.5 and 3) in the particularly

small (< 100 nm) and particularly large particles (> 1000 nm) are striking. As discussed above, such behavior could be caused by an aerosol matrix that is also chiral, meaning that, for example, the E1 enantiomer is preferentially incorporated into the particle phase. Although such behavior can certainly not be ruled out by our measurements, there are other possible explanations. In principle, the particle size is also related to the different ages of the aerosol particles. The very small particle fractions are comparatively young because after particle formation (whether released as small primary particles or newly

formed by gas-particle conversion) condensation processes from the gas phase (or coagulation) cause the particles to grow so that they transition to larger particle regions with increasing age. Thus, a different ratio of the two enantiomers could be explained by the fact that the chiral precursors of the particles formed in the immediate vicinity of the measurement site have a different ratio here than in a more spacious environment. Indeed, Zannoni et al. (2020) was able to measure higher (+)-a-pinene concentrations (i.e., the precursor of E1) closer to the measurement site than far above the ground. However, the

again relatively small E2 / E1 ratios of particles larger than 1000 nm do not fit such an interpretation. Finally, the cause could be an analytical artifact that simply shows small but constant additive effects at lower concentrations (e.g., due to additive cross-interference from coeluting substances). Future measurements will have to show which of the possible explanations really explains the observations made.

Figure 6 again compares the results at the two different sites and the different measurement heights with respect to the

relative abundances of E1 and E2 for all seasons. Unlike the ATTO tower, the triangular mast is fully integrated into the forest structure without any clearing and therefore may better represent the local emission signature at low altitudes. Zannoni et al. (2020) suggested that insects could be a potential source to alter the chiral ratio of monoterpenes. In particular, termites have shown (+)-α-pinene enriched emissions. Apparently, their contribution can shift the chiral ratio from locally dominant (+)-α-pinene to (−)-α-pinene.

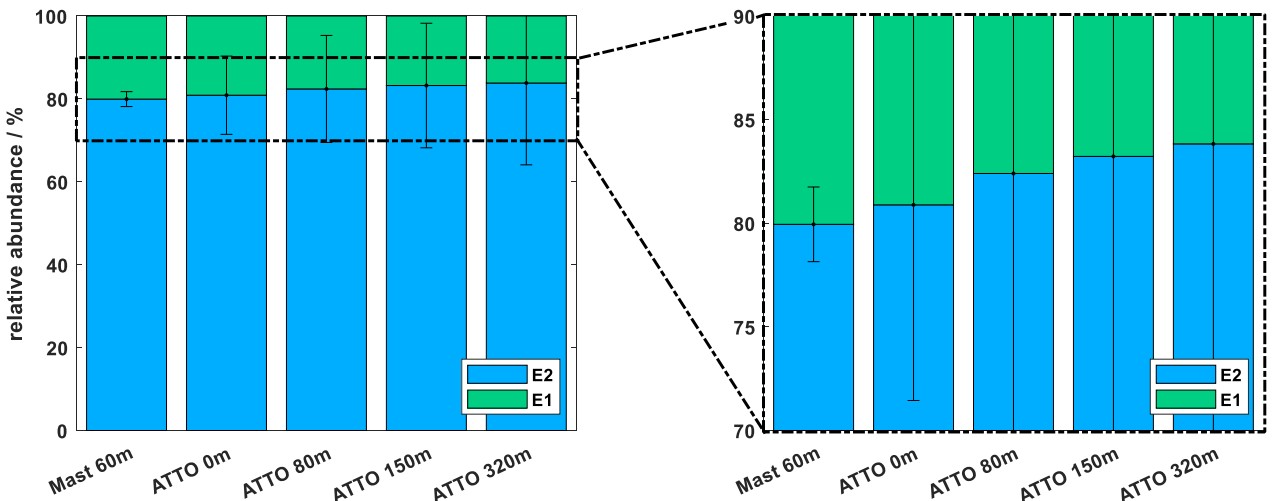


**Figure 6: The average relative abundance is illustrated for five different sampling locations at the triangular mast and ATTO tower for all seasons. The right panel shows an enlarged version to highlight the increasing fraction of E2 with tower height. Apparently, the triangular mast exhibits a different local emission signature caused by the dense inclusion into the forest structure.**

**4 Conclusion**

The present study reports the concentrations and ratios of the two enantiomers of pinic acid within and above the Amazon rainforest at the ATTO site. In all samples, the enantiomer from the oxidation of (−)-α-pinene dominated, confirming the



results of precursor measurements at the same site. Similar to the gas-phase, a gradient in the ratios of the two enantiomers with increasing tower height was also evident in the particle phase. This observation was consistent over three measurement

campaigns from October 2018 to September 2019. The measurements shown at different heights suggest that the results reflect differences in the releases of the chiral precursors, which do depend on the effective measurement height. Thus, local sources influence the results at lower levels and regional sources influence those at higher levels. Somewhat different patterns were observed at 80 m sampling height than at the higher levels, which again indicates that for measurements at 80 m the local emission conditions have a greater influence on the results. The results obtained from the triangular mast

support this assumption.

Overall, the presented results show that the chiral relationship of the biogenic precursor compound α-pinene is preserved in the oxidation products studied here and can thus be used to interpret the biogenic emission sources. However, due to the different lifetimes of the precursor VOC and the particle-bound oxidation products, using the chiral ratios of the longer-lived compound provides a larger-scale picture of precursor emissions, while also revealing local and regional influences.

However, it should also be mentioned that further improvements in the analytical methods would be useful. The molecular formula $C_9H_{14}O_4$ of pinic acid can be formed by a number of other monoterpenes, making reliable quantification at very low concentrations difficult. Therefore, additional chamber experiments with other relevant BVOCs would certainly be helpful to identify and assign the additional LC signals observed. This could potentially provide further helpful information on large-scale biogenic VOC releases and their contributions to the atmospheric particulate phase.

## Data availability


The data sets of this study will be available via the ATTO data repository at http://attodata.org/.

## Author contributions

DL conducted the aerosol sampling and the sample preparation. DL analyzed the data and wrote the first manuscript. NZ helped to collect the aerosol samples. DL and LK were responsible for the chamber experiments. LK and CP provided the

SMPS data. The meteorological data were provided by MS. TH supervised the study. All authors contributed to the scientific exchange and read and revised the manuscript.

## Competing interests

The authors declare that they have no conflict of interest.



## Acknowledgments

For the operation of the ATTO site, we acknowledge the support by the German Federal Ministry of Education and Research (BMBF Contract 01LK1602D) and the Brazilian Ministério da Ciência, Technologia e Inovação (MCTI/FINEP) as well as the Amazon State University (UEA), FAPEAM, LBA/INPA, and SDS/CEUC/RDS-Uatumã. We also acknowledge the support of the Max Planck Society and the Instituto Nacional de Pesquisas da Amazonia (INPA).

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
