# Peer review of "Varying chiral ratio of pinic acid enantiomers above the Amazon rainforest"

_Atmospheric Chemistry and Physics, 2021_

## Author Comment (AC1)

We would like to thank the two reviewers for the valuable and constructive comments/suggestions that helped improve our manuscript. We have carefully revised the manuscript accordingly. Below you will find our point-by-point responses. Reviewer comments and suggestions are written in black, responses in blue.

**RC1**: 'Comment on acp-2021-150', Anonymous Referee #1, 01 Apr 2021

The manuscript by Leppa et al. reports chirally specified measurements of secondary organic aerosol (SOA) at the Amazon Tall Tower Observatory (ATTO) at different altitudes during three measurement campaigns at different seasons. The authors support their observation from the field work with that of the lab experiments. The experiments appeared to be well designed, the manuscript is well written, well structured, and the discussed topic is well suited to the ACP journal.

We appreciate the reviewer's assessment of the quality of the manuscript.

I have several minor comments, which I think can improve this manuscript.

Introduction section: Generally ambient pinic acid is linked to VOC such as alpha-pinene, which is mainly emitted by conifer forest. It is not surprising to see such compounds in the boreal forest e.g. somewhere in northern Europe. Can authors write a couple of sentences on the vegetation type at the sampling site that could explain their observation of pinic acid (and their enantiomers)? I realise that the authors indicated additional contribution sources of alpha-pinene from the local fauna, such as termites, but what about the main contribution source? I also realise the authors are citing other publication that describes the sampling site, however; I feel the vegetation type at the sampling point needs to be described in this manuscript as well as it could possibly explain the abundance and prevalence of the certain enantiomers.

The reviewer raises an interesting point here, which is the attempt to specify more precisely the sources of the VOCs released. However, this is a real challenge at a site like ATTO, simply because the Amazon rainforest is an extremely species-rich and complex ecosystem. Nevertheless, we have added a few explanatory sentences and a reference to further literature on the subject.

Lines 49-51: "It has been shown that physical and chemical properties, such as melting point and water solubility, can then be determined by stereochemistry (Katsumoto et al., 2010; Baker et al., 2015; Cash et al., 2016)." I completely support this statement. My question is along this line (but likely needs to be addressed in either experimental or results and discussion sections), has the author considered the effect of the sample preparation on the recovery and observation of one or the other enantiomer in their study (please see my comment on the extraction below)?

As also described in the text, differences in physical and chemical properties exist, but this only applies to the oligomerization products, not the monomeric degradation products of the biogenic VOCs measured here in this work. Therefore, this does not apply to the sample preparation steps performed here (extraction, solvent evaporation losses, etc.), all of which occur in a non-chiral environment. Enantionmer discrimination in the course of sample

preparation and analysis is not to be expected. More on sample preparation and recovery can be found below.

Experimental section: Line 85: "This also ensured sufficient aerosol mass on the filter".Question: Sufficient for what? For LC/MS analysis? Please clarify.

A sufficient aerosol mass is required to identify even low concentrations of SOA components by LC/MS. Due to detection limits and the various steps involved in sample preparation (filter cut, extraction, evaporation, recovery), an aerosol mass of about 500 µg is required for each sample to ensure reliable results.

"One half of each filter was extracted three times with 1.5 mL in a 9:1 mixture of methanol and water (Fisher Scientific, Optima™ grade) on a laboratory shaker for 30 min. The resulting extracts were combined and filtered through 0.2 µm PTFE 105 syringe filters (Carl Roth, Rotilabo® KC94.1) to remove undissolved material. The solvent was then completely evaporated under a gentle N2 stream and 700 µL of a 9:1 mixture of water and acetonitrile (Fisher Scientific, Optima™ grade) was added to the remaining residue."

Question: What is the point for using methanol/water for extraction and then resuspending the extracts in water/acetonitrile. I understand that likely the authors wanted to match their sample solvent with that of mobile phase, but I am not sure if this is clear to the reader. Also, do you expect any methylation of the carboxylic and hydroxylic groups when extracting in methanol? I assume not within the 30 min of the extraction process, but could the authors state whether their samples were evaporated to dryness immediately or stored (e.g. in the fridge ) prior to evaporation step and for how long? Also, does the selected extraction solvent affect solubility of one or the other enantiomer? Was this checked? Does this have any effect on your results and conclusion? Does the evaporation step have any affect on this as well?

The reviewer is correct in his assumption that the resuspension of the extracts in water/acetonitrile is an important prerequisite for a high separation efficiency of the subsequent chromatography. This is now explained to the reader with an appropriate text, since this was in fact only indirectly apparent from the previous text.

The aspect of possible methyl ester formation of carboxylic acids during extraction is also interesting. Indeed, the formation of methyl esters cannot be completely ruled out (Sauerschnig et al., 2018). However, such artefact formation is mainly problematic in the detection of the methyl esters themselves; in the results presented in this paper, ester formation would merely represent a loss of analyte, i.e. reduce the recovery rate. Although this is also an undesirable effect of methanol, it has been shown in sample preparation experiments or by standard addition that such losses are low (numerical value). Furthermore, such losses are unlikely to discriminate both optical isomers, since a stereoselective reaction can be excluded here. However, we discuss this aspect in more detail in the text on sample preparation now.

Question: Besides MS calibration, have you run any system suitability mixtures to assess the target sensitivity and chromatographic performance?

Several characteristic marker species for biogenic SOA, such as pinic acid, pinonic acid, 3-methyl-1,2,3-butanetricarboxylic acid (MBTCA) but also compounds such as levoglucosan, were used to check chromatographic performance. For this purpose, calibration solutions were prepared in a mixture of water and acetonitrile (9:1) with concentrations ranging from 0.5 ng

mL$^{-1}$ to 800 ng mL$^{-1}$. Solutions for blank determination were prepared in the same way. This information can now also be found in the manuscript text.

Results and discussion: Figure 4, it would be useful to add the reasons for the observed gaps e.g. on the 25$^{th}$, 27$^{th}$ of October (plot 'a') to the legend or adding the break lines to the date axis. This also applies to the Figure 4b, c, and d.

The gaps in graphs a, b, c, d are the result of missing filter samples due to weather conditions (e.g., soaking of filters) or infrastructural or instrumental problems (e.g., power failure).

**RC2**: 'Comment on acp-2021-150', Anonymous Referee #2, 13 Apr 2021

The manuscript presents interesting novel results on the enantiomeric ratio of pinic acid in aerosol samples collected in the Amazon forest at different heights. The manuscript is generally well written and presents the results in a straight-forward way. The presentation focuses only on pinic acid concentrations and comparison with previous work by the same research groups, and the manuscript would benefit from comparison with measurements of e.g. ozone during the sampling periods and possibly previous studies of SOA in the region.

We appreciate the reviewer's assessment of the quality of the manuscript.

Specific comments:

Line 44-45: "Although chiral molecules play a critical role in insect and plant communication (Phillips et al., 2003; Mori, 2014), they are rarely distinguished in studies regarding atmospheric chemistry." Even though there are only few studies, a proper overview of previous findings should be presented. This could include the work of Noziere and colleagues on isoprene SOA (tetrols), as well as the review by Cash et al., 2016, which is only very briefly mentioned in the current version of the manuscript.

Nozière, N. J. D. González, A.-K. Borg-Karlson, Y. Pei, J. P. Redeby, R. Krejci, J. Dommen, A. S. H. Prevot and T. Anthonsen, Geophys. Res. Lett., 2011, 38 DOI:10.1029/2011gl047323.

J. Gonzalez, A.-K. Borg-Karlson, P. Artaxo, A. Guenther, R. Krejci, B. Noziere and K. Noone, Environ. Sci.: Processes Impacts, 2014, 16, 1413–1421.

We have taken up this suggestion and now write, " Although there are a number of atmospheric chemical studies that have dealt with the different diastereomers in isoprene oxidation (Nozière et al., 2011; González et al., 2014), the analysis of enantiomers using chiral phases and their direct analysis is rare."

Line 113: Was the extraction efficiency and recovery tested?

The filter extraction method described in Section 2.3 was evaluated to assess extraction efficiency and recovery. For this purpose, blank filters were doped with 50 ng mL$^{-1}$ and 1000 ng mL$^{-1}$ α-pinene oxidation products, i.e., pinic acid and MBTCA, respectively. The experiments were performed in triplicate. Based on linear regression, recoveries of (93.6 ± 3.9) % and (92.5 ± 5.5) % were calculated for pinic acid and MBTCA, respectively.

Line 117: What was the purity of the standard?

The purity of the standard was determined to be above 90% by NMR spectroscopy.

Line 118: I suggest to provide information about quality of standard curves here (correlation coefficients). Furthermore, the method for determination of the limits of detection and the values in ng/m3 should be stated.

Each measurement was calibrated externally by a linear seven-point regression, which yielded calibration functions with correlation coefficients > 0.98. The detection limit was calculated according to DIN 32645 and resulted in an LOD of 1.74 ng mL$^{-1}$ for pinic acid. This information can now also be found in the manuscript text.

Line 210: Which concentrations for ozone and OH were used for the calculation of lifetimes? Relevant levels should be available from simultaneous measurement or previous studies in the region.

We have now integrated the information on assumed OH and ozone concentrations into the text.

Line 213-216: These sentences are unclear. Please clarify.

We have changed the text to better explain the subject matter.

Line 315: Which further improvements in the analytical methods are needed? If you mean the suggestions in the following sentences, I suggest to write this more clearly.

We have clarified this part and added the following explanations: "Therefore, in particular, an increase in the amounts of analyte transferred to LC/MS analysis, e.g., through further sample preparation steps to reduce solvent volumes, would be extremely helpful."

Minor comments:

Pinic acid is written in upper case in the title.

Has been changed accordingly

Figure 6. I suggest to change the order in the label, so it is similar to the figure (upper: E1 Lower: E2).

Although we can understand why the reviewer makes this suggestion, we would prefer to stick with our form of presentation, as the E1 and E2 designations run throughout the manuscript and a representation reversed in Figure 6 could also be misinterpreted.